# Somatic Instability Leading to Mosaicism in Fragile X Syndrome and Associated Disorders: Complex Mechanisms, Diagnostics, and Clinical Relevance

**DOI:** 10.3390/ijms252413681

**Published:** 2024-12-21

**Authors:** Dragana Protic, Roberta Polli, Elisa Bettella, Karen Usdin, Alessandra Murgia, Flora Tassone

**Affiliations:** 1Department of Pharmacology, Clinical Pharmacology and Toxicology, Faculty of Medicine, University of Belgrade, 11000 Belgrade, Serbia; dragana.protic@med.bg.ac.rs; 2Fragile X Clinic, Special Hospital for Cerebral Palsy and Developmental Neurology, 11000 Belgrade, Serbia; 3Department of Women’s and Children’s Health, University of Padova, 35127 Padova, Italy; roberta.polli@unipd.it (R.P.); elisa.bettella@unipd.it (E.B.); 4Pediatric Research Institute Città della Speranza, 35127 Padova, Italy; 5Laboratory of Cell and Molecular Biology, National Institute of Diabetes, Digestive and Kidney Diseases, National Institutes of Health, Bethesda, MD 20892, USA; karenu@niddk.nih.gov; 6Department of Biochemistry and Molecular Medicine, School of Medicine, University of California Davis, Sacramento, CA 95817, USA; 7Medical Investigation of Neurodevelopmental Disorders (MIND) Institute UCDH, University of California Davis, Sacramento, CA 95817, USA

**Keywords:** *FMR1* gene, mosaicism, full mutation, premutation, activation ratio, methylation

## Abstract

Fragile X syndrome (FXS) is a genetic condition caused by the inheritance of alleles with >200 CGG repeats in the 5′ UTR of the fragile X messenger ribonucleoprotein 1 (*FMR1*) gene. These full mutation (FM) alleles are associated with DNA methylation and gene silencing, which result in intellectual disabilities, developmental delays, and social and behavioral issues. Mosaicism for both the size of the CGG repeat tract and the extent of its methylation is commonly observed in individuals with the FM. Mosaicism has also been reported in carriers of premutation (PM) alleles, which have 55–200 CGG repeats. PM alleles confer risk for the fragile X premutation-associated conditions (FXPAC), including FXTAS, FXPOI, and FXAND, conditions thought to be due to the toxic consequences of transcripts containing large CGG-tracts. Unmethylated FM (UFM) alleles are transcriptionally and translationally active. Thus, they produce transcripts with toxic effects. These transcripts do produce some FMRP, the encoded product of the *FMR1* gene, albeit with reduced translational efficiency. As a result, mosaicism can result in a complex clinical presentation. Here, we review the concept of mosaicism in both FXS and in PM carriers, including its potential clinical significance.

## 1. Introduction: Mosaicism

All living things acquire mutations throughout their lives as a result of the failure to properly deal with different types of DNA damage. Thus, all humans are essentially mosaics. This mosaicism can include single nucleotide variants (SNVs), chromosomal abnormalities, as well as changes in the number of repeats at one or more of the many short tandem repeats (STRs) found in human genomes. In addition to DNA sequence changes, variability in the extent of epigenetic modifications, like DNA methylation and histone acetylation, can also occur. Furthermore, X chromosome inactivation (XCI)—the random inactivation of either the paternal or the maternal X chromosome, which takes place in the early female embryo—makes all females mosaics by virtue of having cells in which only the maternal or only the paternal X chromosome is active [1].

The Repeat Expansion Diseases (REDs) are a group of 40+ neurodegenerative or neurodevelopmental conditions that result from expansions of a short tandem repeat (STR) or microsatellite [2]. However, recent evidence has shown that in some of these diseases, changes in repeat number occurring in somatic cells during an individual’s lifetime can significantly affect the age at onset and disease severity [3]. These changes in repeat number are primarily increases or expansions, but somatic contractions are not uncommon, particularly where the repeat numbers are large. These changes occur at different rates in different cells, and as a result, different tissues or cells within such individuals can be mosaic for a range of repeat sizes. In addition, some of these repeats are associated with epigenetic changes, including DNA hypermethylation, that affect the level of expression of the affected gene, and these epigenetic changes can also vary between individuals and within the cells of those individuals. Furthermore, for those diseases that are X-linked, X chromosome inactivation—the process by which one of the two X chromosomes are inactivated during early embryogenesis—can contribute to the mosaicism in females.

Among the REDs, the Fragile X-related disorders, which result from expansion of a CGG repeat tract in the X-linked Fragile X messenger ribonucleoprotein 1 (*FMR1*) gene, represent a particularly interesting group of diseases in which all of these forms of mosaicism are seen. In this review, we aim to present our current knowledge of the extent of this mosaicism and its many clinical implications.

## 2. The Intrinsic Mosaicism of the Fragile X Disorders

The instability of a stretch of CGG tandemly repeated elements within the 5′ UTR of the *FMR1* gene, located on the X chromosome, leads to the family of Fragile X-related conditions. Alleles with >200 CGG repeats are referred to as full mutation (FM) alleles. Such alleles result in FXS, the most common inherited cause of intellectual disability and monogenic form of autism. FM alleles are aberrantly heterochromatinized. This results in transcriptional silencing and the loss of the *FMR1* gene product, FMRP, an RNA-binding protein important for synaptic plasticity and memory [4,5]. Thus, the central pathogenic mechanism in FXS is the lack of FMRP.

*FMR1* alleles with 55–200 CGG repeats are known as premutation (PM) alleles. Such alleles confer a risk of developing Fragile X premutation-associated conditions (FXPAC), which include Fragile X-associated tremor/ataxia syndrome (FXTAS), Fragile X-associated primary ovarian insufficiency (FXPOI), and the Fragile X-associated neurodevelopmental disorders (FXAND). In addition, carriers of a PM are at higher risk of various medical conditions, such as thyroid disorders, neuropathy, fibromyalgia, immune-mediated diseases, migraines, and sleep disturbances [6,7,8].

The molecular basis of the pathology seen in these disorders is still not completely understood. However, in contrast to the silencing of FM alleles, most PM alleles are hyperexpressed, and the resultant elevated levels of the CGG-expanded *FMR1* mRNA are thought to have deleterious effects, perhaps via some combination of the sequestration of rCGG-repeat-binding proteins [9,10], altered and chronic activation of the DNA-damage response [11,12], and repeat-associated non-AUG (RAN)-initiated translation.

Multiple types of mosaicism are seen in both FM and carriers of a PM, including mosaicism for repeat size and DNA methylation. Chromosomal abnormalities, including copy number variations and aneuploidy, can also contribute to mosaicism, and because *FMR1* is X-linked, mosaicism due to XCI is also observed in females.

### 2.1. Size Mosaicism

Alleles containing large numbers of repeats are at risk of expansion in the germline as well as in somatic cells. In general, the risk of expansion is directly related to the trinucleotide repeat number. In the case of the FXDs, expansion risk is affected by the presence of AGG interruptions that are sometimes seen at the 5′ end of the CGG repeat tract. These interruptions reduce both the risk of intergenerational transmission of FXS [13,14] and the extent of somatic expansion [15,16,17,18,19].

Work in mouse and cell models of the FXDs, as well as the study of single nucleotide polymorphisms in women with the PM and in patients with other REDs, has implicated mismatch repair (MMR) proteins in repeat expansion in somatic cells (see [20] for recent review). In particular, MMR factors like MutSβ and MutLγ are drivers of expansion, while other factors, like FAN1, protect against it. Since some of the most expansion-prone cells are post-mitotic neurons, these events do not require cell division. However, expansion is limited to the active X chromosome in women, suggesting that transcription is required [21,22]. Emerging evidence suggests that during transcription of the repeat tract, misalignment of the individual strands of the repeats occurs. This results in the generation of a loop-out on each strand. The combined action of MutSβ and MutLγ results in the production of nicks on the strand opposite each loop-out. Repair of these nicks via the action of Polδ and DNA ligase I are then thought to result in expansions [23]. Over time, these events produce a characteristic change in the allele profile such that not only does the modal allele size increase, but the distribution of alleles about this mode also increases. Since expansion is only seen when the PM allele is on the active X chromosome, expansions in women give rise to a characteristic allele profile in which low levels of expansion result in the appearance of a “shoulder” to the PCR profile of the PM allele. More extensive expansions result in the emergence of a distinct allele population [22]. The difference between the mode of the smaller allele population and the larger one represents the average number of repeats added during the woman’s lifetime. The change in the distribution of the expanded allele population allows the extent of expansion to be estimated in the absence of knowledge of the size of the original inherited allele. Somatically stable alleles have a narrow distribution that changes little with time, while unstable alleles become broader with time [22]. Such expansion profiles are consistent with expansions occurring in almost all cells in a given cell population, with the addition of a small but slightly variable increase in repeat number with each expansion event. However, these events may be very frequent in some cell types, particularly when the repeat number is large. As a result, large changes in repeat number can accumulate over an individual’s lifetime. However, for reasons that may be related to some combination of the expression of factors that modulate expansion and the transcriptional activity of the gene, cell-specific differences in expansion rates are seen. For example, PBMCs tend to show less expansion than other cell types, including neurons, and thus the allele sizes in blood may not necessarily reflect the allele distribution in more disease-relevant cell types.

Contractions are also seen, although much less is known about the mechanisms involved. Small contractions are seen in embryonic stem cells from an FXD mouse model when MMR factors are missing [24], consistent with the failure to repair strand-slippage products arising during replication. However, many contractions involve the loss of a large and variable number of repeats and occur in a MMR-dependent fashion [4]. Such expansions are apparent when large epigenetically silenced alleles are reactivated, implicating transcription in this process. However, some contractions are also seen in individuals who have large epigenetically silenced alleles, suggesting that a subset of contractions may not be transcription-dependent. Contractions that result in the repeat number falling below a certain critical threshold for expansion may be stably propagated. However, if contractions remain above this threshold, subsequent expansion may still occur, resulting in more than one allele in the population that is increasing in length over time.

Partial or full *FMR1* gene deletions have also been reported, and if this occurs post-zygotically, this too would result in somatic mosaicism [17,25,26,27,28,29,30,31,32,33,34]. Chi-like recombinogenic elements flanking the *FMR1* repeat region have been suggested as one source of these large deletions [32], although it is possible that contractions originating in the repeat are the trigger for these events.

While expansion mechanisms that give rise to large changes in repeat number in a single step have been reported in yeast [35], all expansions observed thus far in a mouse model are small and MMR-dependent [21], and whether large expansions occur in humans is unclear. Thus, it is likely that in cells containing alleles with very different sizes, the smaller alleles arise by contraction from longer ones.

### 2.2. Methylation Mosaicism

Hypermethylation of the 5′ end of *FMR1* gene is triggered when the CGG repeat number exceeds a threshold of ~200 repeats [36]. This is thought to occur early in embryogenesis, although exactly when remains the subject of some debate [37,38,39,40,41,42,43,44]. In addition to CpG methylation, silenced alleles are associated with histone modifications characteristic of both facultative and constitutive heterochromatin, including demethylation and trimethylation of H3K9, trimethylation of H3K27, and deacetylated H4K16 (reviewed in [45]). Consistent with this, treatment of patient cells with 5-aza-2′-deoxycytidine (AZA), a DNA methyltransferase inhibitor, can partially restore *FMR1* gene expression [46,47,48,49], as can targeting the ten-eleven translocation (TET) family of proteins (TETs) to the *FMR1* locus [50,51]. Further, it has been demonstrated that ttargeted demethylation of the *FMR1* CGG repeats by dCas9-Tet1 reactivates gene expression in Fragile X cells [51]. These studies suggest that using TET proteins or CRISPR-Cas9 systems to target the *FMR1* promoter and ultimately reverse gene inactivation through epigenome editing could represent a potential therapeutic strategy for FXS. Finally, treatment with H3K9 and H3K27 inhibitors does not reactivate the gene, although they do delay the silencing seen in AZA-treated cells when the AZA is withdrawn [49,52]. This has been interpreted to mean that DNA methylation occurs downstream of these modifications and, once established, is perpetuated. However, inhibitors of H4K16 deacetylation are able to increase *FMR1* expression even in the absence of AZA, suggesting that deacetylation may be a late step in the silencing process [53].

For reasons that are not well understood, some FM carriers escape methylation completely or partially [54,55,56,57,58,59]. Methylation of alleles in the PM range are also occasionally seen [60]. This seems to occur mostly in the upper PM range and may reflect either differences in the threshold for methylation in different individuals or their origin from contraction of methylated FM alleles [59,60].

### 2.3. Somatic Mosaicism in FXS

Almost all full mutation carriers (FM, Figure 1a,b) are characterized by an array of alleles of different sizes in different cells, likely arising from contractions of a large inherited allele. However, since fully methylated FM alleles produce no FMRP no matter their size, the use of the term “size mosaicism” is usually limited to those individuals who also have unmethylated alleles. Since such alleles would generate not only some FMRP but also FMR1 transcripts that would be deleterious [61,62,63,64], thus complicating the clinical picture. Historically, the term FM size mosaicism has been used to describe individuals with hypermethylated full mutation allele that also have a single unmethylated allele. This unmethylated allele may fall within the PM range (Figure 1a,c), the intermediate (44–54 CGG), or even normal size allele (<44 CGG). When FXS individuals carry different sizes of the repeat expansion in different cells, including some methylated FM alleles and a series of unmethylated alleles within the PM and FM range, they are referred to as FM size/methylation mosaics (Figure 1a,d).

Both intra-tissue and inter-tissue size and methylation mosaicism have been reported in both FM and PM carriers [22,56,60,64,69]. However, given cell-type differences in the extent of somatic instability, the degree of size and/or methylation mosaicism detected in blood may not necessarily reflect the mutation pattern present in brain cells or in cells of other tissues (Figure 2). Thus, the clinical consequences may not always be fully predicted from what is detected in blood samples [34,64].

Allelic instability and somatic mosaicism have also been reported in both males and females with a PM and are associated with some PM phenotypes (Figure 3) [22,60]. It has been suggested that both cis- and trans-acting genetics can lead to somatic instability in females with a premutation. Specifically, in addition to the CGG repeat length and the presence of AGG interruptions, two DNA repair genes, *FAN1* and *MSH3*, appear to play a role in the risk of somatic expansion [22], as also reported in other repeat expansion disorders. In addition, somatic instability proneness to somatic expansion is associated with the presence of ADHD in women with the premutation [70], suggesting a role of the CGG expansion in the clinical premutation phenotypes and its potential to affect disease penetrance, age at onset, and disease severity. The amplitude of these phenomena across tissues could be very crucial in contributing to the variable penetrance of FXPAC pathology seen in carriers of the premutation. Comparably, the role of these genetic factors has also been reported in the FXD mouse model [71,72]. Thus, a better understanding of the full range of genetic factors affecting expansion risk may contribute to better assessments of disease risk in PM carriers, as well as the risk of transmission of FXS.

### 2.4. X Chromosome Mosaicism

In addition to size and methylation mosaicism, because of XCI, females have an additional source of mosaicism since they express the expanded allele only in a fraction of their cells. Although there is no evidence of repeat-related skewing of the X chromosome, normal XCI can result in a wide variation in the fraction of cells in which the active X chromosome contains the expanded allele. The *FMR1* gene is subject to X-inactivation in the somatic cells of females, and the activation ratio (AR; Figure 4) represents the fraction of the normal allele carried on the active X chromosome. In principle, high activation ratios would be associated with normal levels of *FMR1* mRNA and FMRP in a higher proportion of cells. This explains why females with either an FM or a PM generally present with a milder phenotype and less disease symptoms than males [73].

### 2.5. Chromosomal Abnormalities

FM alleles exhibit a folate-sensitive fragile site that gives this group of diseases their name. This fragility is thought to arise because of difficulties with the replication of the long CGG repeat tract [73,74]. As a result, during times of replication stress, a form of mitotic DNA synthesis (MiDAS) is deployed to try to ensure that replication of the affected X chromosome is complete before cell division occurs [74]. This process is thought to involve a form of break-induced replication (BIR) that is dependent on Rad51 and SLX1. BIR is error-prone due in part to higher-than-normal rates of strand switching. This could account for the duplication of regions of the X chromosome in the vicinity of the repeat, which is often seen in patient cells exposed to folate stress [75]. Stalled replication forks that are not resolved before cell division may result in chromosome mis-segregation. This phenomenon may account for the observation that ~5% of FM female fetuses are mosaic for the loss of the affected X chromosome, a condition known as Turner syndrome (TS) [76].

To further complicate the picture, in some cases, the presence of extra normal alleles in the normal CGG repeat range is observed and may reflect either X chromosome aneuploidy or the products of repeat contraction [15,77,78,79,80].

**Figure 4 ijms-25-13681-f004:**
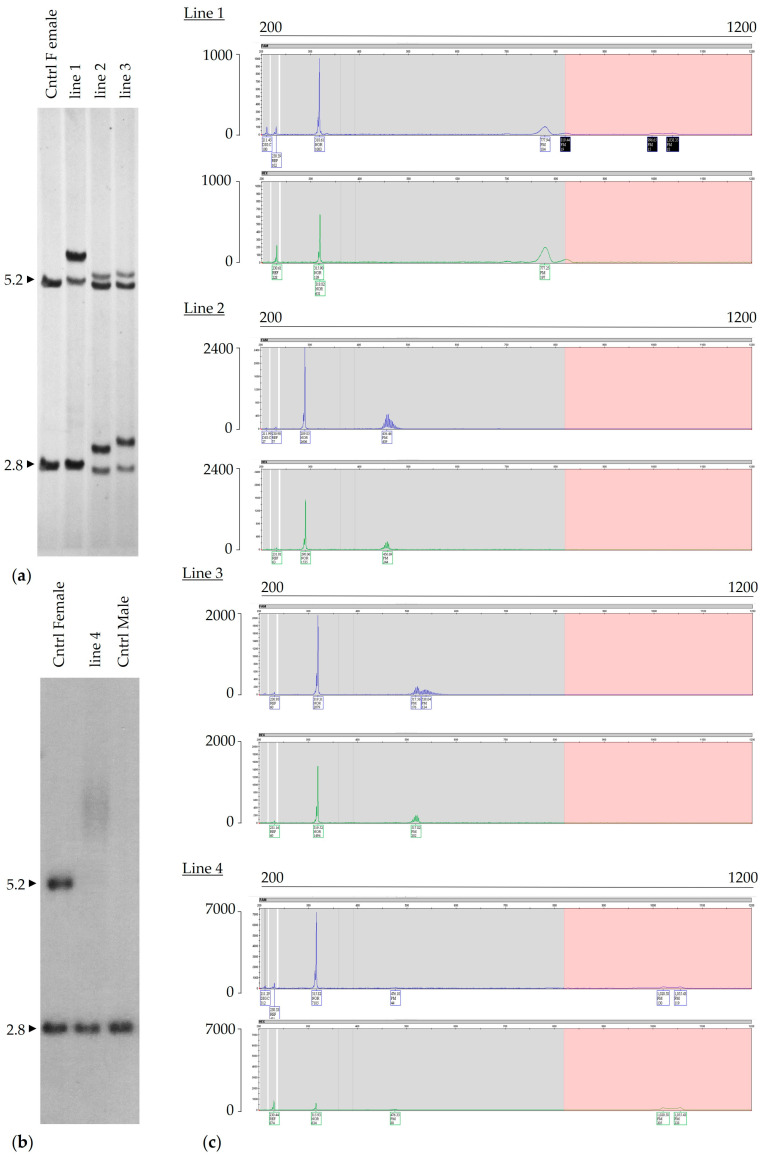
Activation ratio. Southern blot analyses (panels **a**,**b**) and capillary electropherograms (**c**) of PCR products (triplet repeat PCR with/without the CGG primer and methylation PCR) from three females with a premutation: one with a complete skewed AR (line1), and the other two with an AR of 0.41 and 0.38, respectively (lines 2 and 3). One female with a full mutation and a very skewed AR of approximately 0.85 is shown in lane 4. Measurements of methylation status and activation ratio (AR) are shown as previously described [81]. For Southern blot analysis, DNA was digested with EcoRI/NruI and probed as described in [65]. MW = 1 kb size ladder marker; Cntrl F = normal female, negative control. M = normal male. In the electropherograms plots, the x-axis indicates the number of base pairs, and the y-axis indicates the relative fluorescence intensity.

## 3. Biological Effects of Somatic Mosaicism

Several hypotheses are emerging in the literature on the possible pathways contributing to the different forms of FXD as driving pathogenic mechanisms. For example, it has been shown recently that mitochondria can modulate cell death in the FXDs. A recent study indicated that decreased FMRP expression may play a crucial role in mitochondrial function, suggesting that the lack of FMRP induces mitochondria that are uncoupled due to an inner membrane leak, reducing efficiency by consuming fuel rather than conserving energy. Such mitochondria in the brains of *Fmr1^−/y^* mice exhibit impaired thermogenic respiration caused by a coenzyme Q-regulated proton leak, leading to synaptic spine abnormalities and behavioral deficits [82]. The same group of researchers described that the decreased FMRP expression in fragile X neurons has been linked to a defect in ATP synthase that causes a leak. Inhibiting this leak has been shown to restore normal cellular function and mitigate behavioral disease phenotypes [83]. A study on fibroblasts derived from unaffected control males, FXS patients, individuals with premutation (PM), and unmethylated full mutation (FM) revealed that decreased FMRP expression might be responsible for altered mitochondrial morphology (“donut-shaped mitochondrial morphology”), upregulation of mitochondrial oxidative phosphorylation complexes and other proteins, and increased cellular sensitivity to apoptotic stimuli [84]. All listed biological mechanisms at the cellular level might explain the clinical phenotypes of FXD.

For transcriptionally active alleles, there is a direct relationship between repeat number and *FMR1* transcript levels [85]. Consistent with this, carriers of active PM alleles show a direct relationship between repeat number and onset of FXTAS symptoms [86,87]. The presence of large, unmethylated alleles, whether PM or FM alleles, would thus confer a risk of pathological effects related to the toxic effect of the *FMR1* mRNA. In contrast, FMRP expression levels are inversely correlated with the number of CGG repeats, a phenomenon thought to be related to decreased translation efficiency and stalling of the 40S ribosomal subunits en route to the start of translation [88,89]. As a result, individuals with *FMR1* expanded unmethylated alleles, even in the upper PM range, show decreased FMRP expression [89,90,91,92]. Since FMRP is important for synaptic function (reviewed in [93]), such alleles may impact learning and memory. Methylated alleles, whatever their size, would make little or no *FMR1* mRNA or FMRP. Such alleles would thus not convey a risk of RNA-related toxicity but could have consequences for cognitive function.

Thus, size and methylation mosaicism may contribute to the observed heterogeneity of the clinical phenotype in PM and FM carriers, including variation in cognitive abilities and behavioral phenotypes [60,63]. Mosaic individuals, in general, tend to have better intellectual functioning and fewer FXS symptoms compared to those with fully hypermethylated FM alleles [55,63,94,95,96,97,98,99,100,101,102,103,104]. For example, a high percentage of methylation (>80%) has been reported to directly correlate with a decreased *FMR1* mRNA and FMRP expression levels and inversely correlate with full-scale IQ scores, while a lower percentage of methylation (<80%) correlates with higher cognitive abilities [101]. Pretto et al. (2014) observed a higher number of co-occurring clinical features (IQ, anxiety, ADHD, perseveration, tantrum, ASD) in young males with CGG repeat size in the upper PM range and/or crossing the FM threshold, and with methylation mosaicism (<2% up to 48%) [96]. Several FXTAS mosaic male cases, diagnosed based on clinical assessments, magnetic resonance imaging data, and the presence of intranuclear inclusion (the pathological hallmark of FXTAS), have been reported as having FM methylation or size mosaicism [105,106,107,108,109,110]. Additionally, a FORWARD (the Fragile X Online Registry with Accessible Research Database) study reported better cognitive functioning, adaptive behaviour, and less social impairment in individuals with methylation mosaicism versus size mosaicism [111]. Further, numerous studies have highlighted the significant impact of methylation status and AR on the neurocognitive and physical phenotypes in females with the FM [112,113,114]. Recently, the correlation between AR and clinical outcomes was also reported in female PM carriers, where a notable association was observed between higher AR and better performance, verbal, and full-scale IQ scores [81]. Likewise, a reduced likelihood of experiencing depression and the occurrence of medical conditions was correlated with a higher AR [81]. Several other studies have also suggested an association between AR values and cognitive and behavioural challenges in PM carriers [115,116,117,118,119].

## 4. Concluding Remarks

Traditionally our understanding of the FXDs has been based on the idea that PM symptoms were a simple function of the inheritance of a large, transcriptionally active *FMR1* allele, and that the symptoms displayed by individuals identified as FM carriers resulted simply from the consequences of the repeat-induced transcriptional silencing of the *FMR1* allele. However, recent findings suggest a more complex clinical picture in which different kinds of mutational profiles, including various combinations of different allele sizes and percentage of methylation, all falling under the umbrella of somatic mosaicism, can result not only in variations in symptom severity but also, in some cases, in blurring of the distinction between FM and PM carriers.

To note, properly accounting for somatic mosaicism is not simple; firstly, expanded alleles are challenging to analyze, whether by PCR, long-read sequencing, or Southern blot, with size mosaicism adding to this difficulty. Furthermore, size mosaicism adds to the challenge of evaluating methylation mosaicism. Finally, it is important to bear in mind that while most genetic analysis is based on the use of peripheral DNA sources like peripheral blood leukocytes, buccal cells, or fibroblasts, it is hard to know how closely the mosaicism in these cell types mirrors what is present in the most biologically relevant cell types, like neurons and oocytes.

Nevertheless, an appreciation of the causes and consequences of the different forms of somatic mosaicism in FXDs, their correct detection in diagnostic testing, and inclusion and definition in clinical reports is important not only to improve the diagnosis and treatment of FXDs but also for the interpretation of the results of clinical trials and thus the evaluation of therapeutics to treat these diseases.

## Figures and Tables

**Figure 1 ijms-25-13681-f001:**
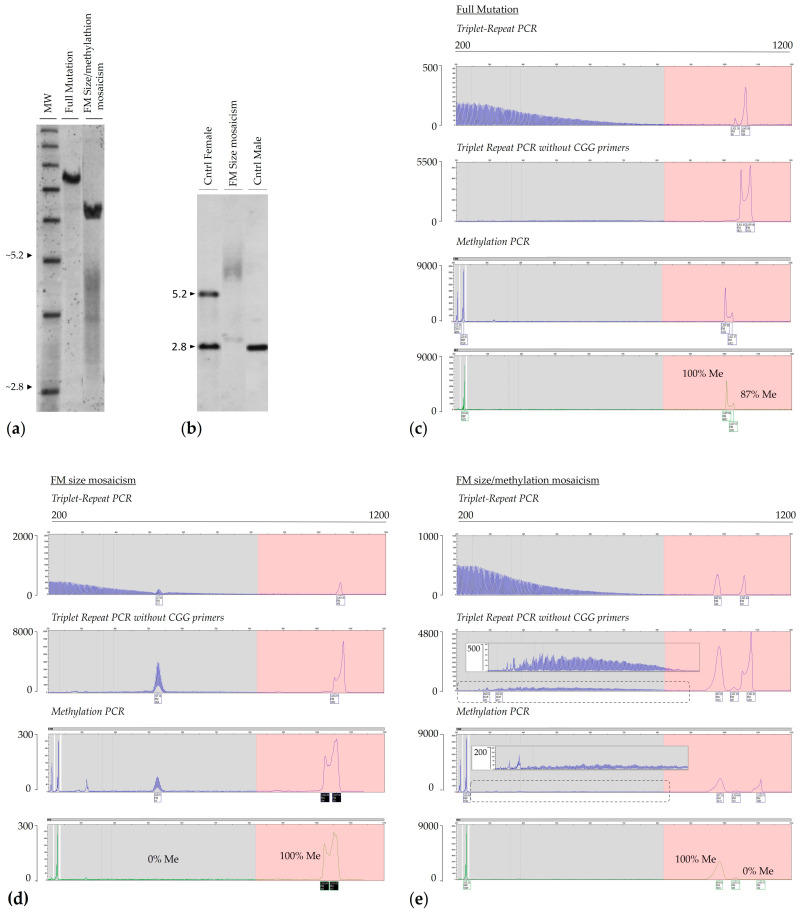
Representative examples of *FMR1* mosaic categories. Southern blot analysis (**a**,**b**) shows the presence of non-methylated and methylated *FMR1* alleles in DNA isolated from peripheral blood derived from three males: full mutation, 100% methylated (**a**, line 2); full mutation size mosaic (**b**, line 2); and full mutation size/methylation mosaic (**a**, line 3). 1 kb molecular weight size marker (MW, **a** line 1); female and male negative controls (**b**, lines 1 and 3). The normal unmethylated (2.8 kb) and a normal methylated (5.2 kb) allele size are shown on the left (**b**). Southern blot analysis was carried out on genomic DNA digested with EcoRI and NruI and probed with the specific *FMR1* genomic probe StB12.3, as described in [65]. Corresponding electropherograms of triplet repeat PCR with/without the CGG primer and methylation PCR from peripheral blood DNA of (**c**) a full mutation male [(**a**) Southern blot lane 2], of (**d**) a full mutation size mosaic male [(**b**) Southern blot lane 2], and (**e**) a full mutation size/methylation mosaic male [(**a**) Southern blot lane 3] are shown. Southern blot and PCR analysis of a male with a full mutation (**c**) show the presence of a methylated FM greater than 200 CGG repeats. Southern blot analysis of a full mutation size mosaic male (**d**) shows PM fragments above 5.2 kb, indicating the presence of several *FMR1* methylated alleles greater than 200 CGG in size and of an allele in the premutation range. Triplet repeats PCR analysis with/without the CGG primer electropherograms shows the presence of a premutation allele with 100 CGG and FM (>200 CGG) alleles. mPCR reveals that the PM allele of 100 CGG was unmethylated, and the FM alleles were completely methylated. Southern blot analysis of a full mutation size/methylation mosaic male (**e**) shows the presence of a continuum of unmethylated alleles ranging approximately from 90 to 750 CGG repeats (~70% unmethylation) and the presence of hypermethylated alleles greater than 200 CGG repeats (above 5.2 kb, 30% methylation). The triplet repeats PCR with/without the CGG primer and mPCR electropherograms show the presence of a broad range of unmethylated alleles ranging from the normal to the low full mutation range. PCR and methylation PCR were carried out as reported in [66,67]. In the electropherogram plots, the x-axis indicates the number of base pairs, and the y-axis indicates the relative fluorescence intensity. Methylation percentage was calculated as previously described in [68].

**Figure 2 ijms-25-13681-f002:**
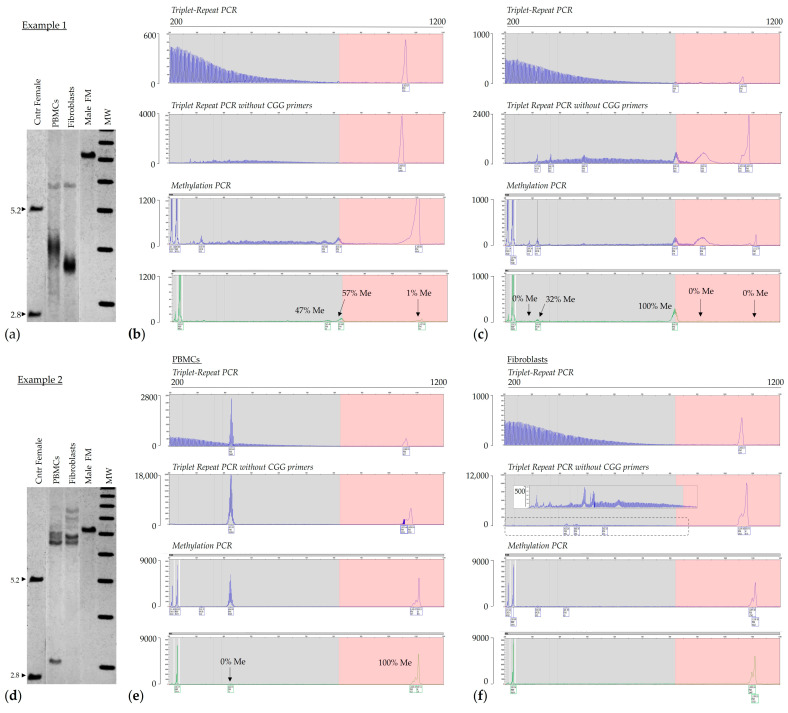
Inter- and intra-tissue mosaicism. Southern blot analyses (SB; **a**,**d**) and electropherograms of PCR products (triplet repeat PCR with/without the CGG primer and methylation PCR) from DNA isolated from peripheral blood (panels **b**,**e**) and primary fibroblast cells (panels **c**,**f**) of two examples of FM mosaic males (examples 1 and 2) are shown. Both SB and PCR analyses show CGG instability, as illustrated by the presence of highly heterogeneous fragments across all amplification ranges and differences in *FMR1* methylation when comparing blood to primary cultured fibroblasts. For SB analysis, DNA was digested with EcoRI and NruI and probed as described in [65]. MW = 1 kb size ladder marker; Cntr Female = normal female control. Male FM = full mutation male, positive control. In the electropherogram plots, the x-axis indicates the number of base pairs, and the y-axis indicates the relative fluorescence intensity. Methylation percentage was calculated as described in [68].

**Figure 3 ijms-25-13681-f003:**
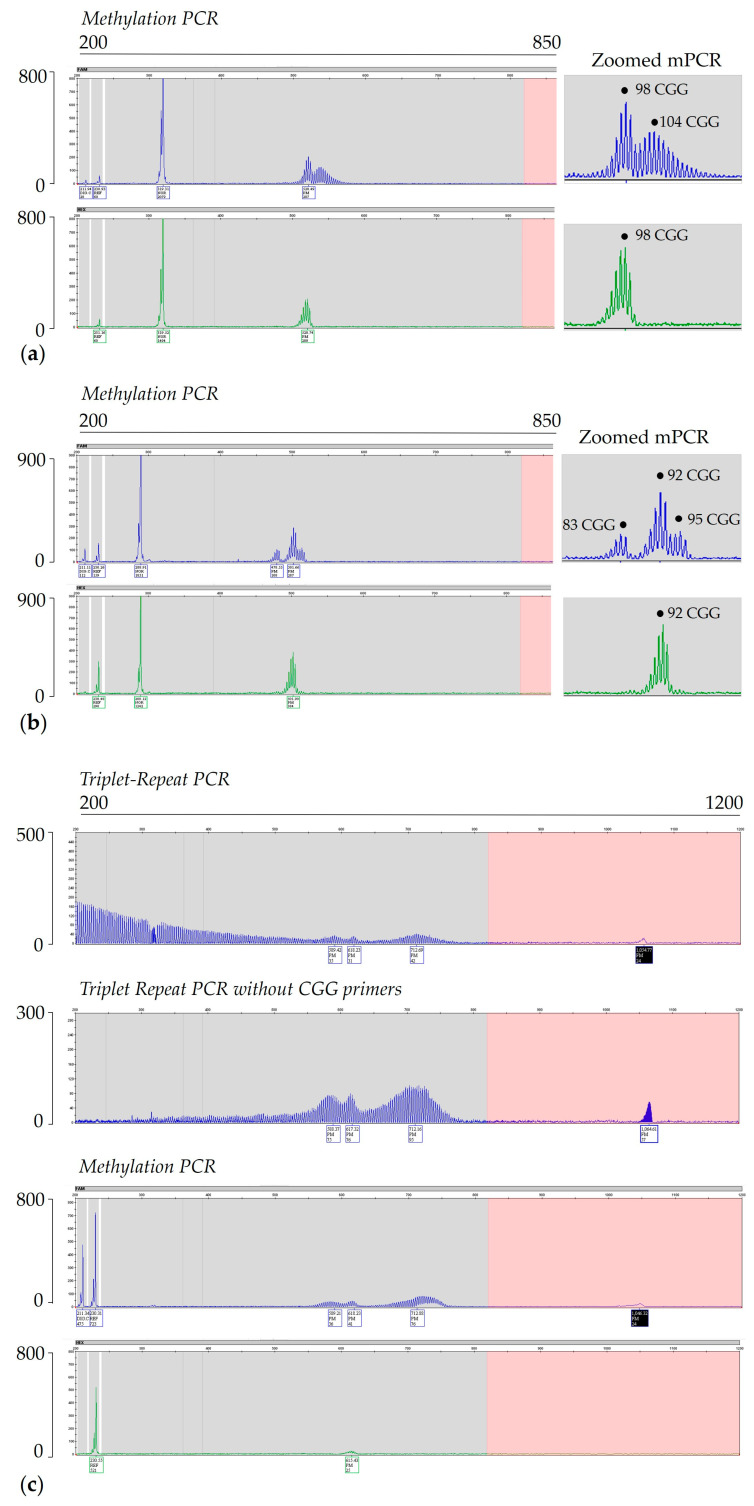
Somatic allelic instability in PM carriers. Panels (**a**,**b**) show examples of methylation PCR profiles observed in two female PM carriers. Capillary electropherograms show increasing levels of somatic expansion. (**c**) Example of triplet repeats PCR with/without the CGG primer and mPCR in a male with somatic instability (FM size/methylation mosaic), characterized by the presence of FM alleles and a wide range of premutation alleles with a complete absence of methylation. The x-axis indicates the number of base pairs, and the y-axis indicates relative fluorescence intensity.

## Data Availability

Data generated by this study will be available upon request.

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
