# Peer review of "Somatic Instability Leading to Mosaicism in Fragile X Syndrome and Associated Disorders: Complex Mechanisms, Diagnostics, and Clinical Relevance"

_ijms, 2024, doi:10.3390/ijms252413681_

Round 1
Reviewer 1 Report
Comments and Suggestions for Authors
This review article examines the diverse factors underlying the mosaic expression of fragile X syndrome and related disorders caused by CGG expansions in the FMR1 gene, including FXTAS, FXPOI, and FXAND. The authors highlight two key mechanisms—somatic repeat instability and DNA methylation—presenting these topics in a comprehensive yet accessible manner with clear and straightforward language, emphasizing the clinical impact of mosaicism on the wide variability and overlapping phenotypes within and across these diseases. This discussion is particularly valuable for assisting clinicians in assessing disease severity and developing personalized therapeutic strategies. Consequently, the article warrants publication.
To further enhance the manuscript's impact and ensure its suitability for publication, I recommend the authors expand on the following areas:
a. Repeat Instability
- Extend on how do DNA replication, repair, and transcription processes are thought to contribute to this phenomenon?
- At what developmental stages and in which tissues are these mechanisms most active?
- What do we know regarding repeat instability in the brain and other tissues relevant to disease pathology?
b. DNA Methylation and Epigenetic Modifications
- When does FMR1 hypermethylation occur, and how is it maintained and/or lost? Are these processes mediated by active or passive mechanisms?
- What is known about FMR1 de-methylation in the brain, particularly concerning high TET activity?
- how do FMR1 methylation changes impact the cellular phenotype, including RNA toxicity and RAN-translation?
c. Clinical Translation
- Expand on how the findings discussed in this review could inform patient care. For example, how precision medicine approaches be leveraged to improve treatment strategies?
Addressing these points will significantly strengthen the manuscript, providing deeper insights into disease mechanisms and their implications for clinical practice.
Typo correction: Remove the first word ("In") in the abstract.
Author Response
Dear Reviewer,
Thank you for your comments. We accepted all your suggestions and accordingly we have completely rewritten and improved our manuscript. Following your recommendations, we have expanded on the following areas: (a) Repeat Instability, (b) DNA Methylation and Epigenetic Modifications, and (c) Clinical Translation. However, the changes are not marked using Track Changes as originally expected, because there was an initial idea to resubmit this article as a new submission under a new ID number (as initially suggested by Editorial Office). We are confident that you will be able to follow the article’s improvements.
Thank you.
Reviewer 2 Report
Comments and Suggestions for Authors
Comments and Suggestions for Authors
This review, is focused on the concept of mosaicism and describes the possible clinical conditions that are due to genetic and epigenetic modifications in both FXS and in carriers of a PM. The authors add a correlation to the clinical phenotypes and comment on the potential biological significance. Overall, I find that this review is well written but limited to readers with specific genetic interests. Moreover, the biological mechanisms that are behind the clinical phenotypes that are described in this review should be better analyzed in the dedicated paragraph.
Specific comments and suggested changes
1. Figures
The Figures are too small and need a general reorganization to become readable and clear. The Authors are recommended to change them in order to reach a sufficient standard for publication. Figure 2 misses the panels a, b and c.
2. Text
The text contains small mistakes that should be fixed before publication:
- Page 1, the last Author of the Author’s line is missing.
- Please change at the beginning of the abstract “In” with “The”
- All along the text the gene names should be in italics, please replace them
- Page 2, the abbreviation for FMR1 inserted at the beginning of paragraph 2, should be anticipated at the end of paragraph 1, where it is firstly named.
- Page 6, some references are quoted in the wrong format i.e. Hwang et al., instead of the reference number
3. Contents
The Authors should improve the paragraph 5 by mentioning the biological mechanisms at the cellular level that might explain the clinical phenotypes. Several hypotheses are emerging in the literature on the possible pathways participating to the different forms of this disease as driving pathogenic mechanisms. For example, it has been shown recently that mitochondria can modulate cell death in the FXDs (Licznerski P. et al., 2020 Cell, ATP Synthase c-Subunit Leak Causes Aberrant Cellular Metabolism in Fragile X Syndrome; Grandi M. et al., 2024 Int J Mol Sci, Mitochondrial Dysfunction Causes Cell Death in Patients Affected by Fragile-X-Associated Disorders), a fact which can be easily correlated to damage in specific tissues.
Author Response
Dear Reviewer,
Thank you for your comments. We accepted all your suggestions and accordingly we have completely rewritten and improved our manuscript. Following your recommendations, we have expanded on the following areas: (a) Figures, (b) Text, and (c) Contents. However, the changes are not marked using Track Changes as originally expected, because there was an initial idea to resubmit this article as a new submission under a new ID number (as initially suggested by Editorial Office). We are confident that you will be able to follow the article’s improvements.
Thank you.
Round 2
Reviewer 2 Report
Comments and Suggestions for Authors
The review has been improved according to suggestions, but the figure labels and numbers are still not readable, The Authors should further work on Figure quality.